

# Reduced Cooling in the Norwegian Atlantic Slope Current: Investigating mechanisms of change from 30 years of observations

Till M. Baumann[1,2], Øystein Skagseth[1,2], Randi B. Ingvaldsen[1], and Kjell Arne Mork[1,2]

[1]Institute of Marine Research (IMR), Bergen, Norway
[2]Bjerknes Centre for Climate Research, Bergen, Norway

**Correspondence:** Till M. Baumann (till.baumann@hi.no)

**Abstract.** The Norwegian Atlantic Current (NwAC) is a principal conduit for poleward heat and salt transport within the Atlantic Meridional Overturning Circulation (AMOC) and plays a key role of water mass transformation in the Nordic Seas. Its variability exerts a critical influence on high-latitude climate, Arctic Ocean inflows, and deep-water formation in the Nordic Seas. This study presents a comprehensive analysis of a 30-year (1993–2022) hydrographic dataset from four repeat sections

across the NwAC, spanning from the southern Norwegian Sea (62.8°N) to Bjørnøya (74.5°N). Hydrographic measurements of temperature and salinity, along with derived relative geostrophic velocities, were combined with surface geostrophic currents from satellite altimetry to obtain absolute geostrophic velocities throughout the water column at each section. This allows us to robustly define the current core of the NwAC and assess its properties. The data reveal substantial variability in water properties and transport across seasonal to multi-annual timescales, alongside significant warming trends. While the cooling

and freshening of Atlantic Water (AW) along the Norwegian coast is a persistent feature, our analysis indicates a decreasing cooling trend north of Lofoten (69°N). We examine three potential drivers of this reduced cooling: (1) increased advection speed within the current core, (2) reduced lateral heat loss due to decreasing eddy-activity, and (3) decreased air-sea heat fluxes. We find no evidence for any changes in eddy kinetic energy, but both increased advection speed and reduced air-sea heat loss may contribute to the observed decline in cooling. Simple box model estimates suggest that while neither of the two

factors can explain all variability observed in the cooling north of Lofoten, changed heat fluxes can quantitatively account for the long term trends. Our results imply a northward amplification of AW warming along the northern rim of the Atlantic Overturning Circulation

## 1   Introduction

The Norwegian Atlantic Current (NwAC) is a crucial component of the Atlantic Meridional Overturning Circulation (AMOC)

and serves as the primary conduit for heat and salt transport from the subtropical North Atlantic to the Arctic Ocean. The Atlantic Water (AW) within the current undergoes significant cooling and freshening as it traverses the Nordic Seas, impacting Arctic Ocean inflows, deep-water formation, and high-latitude climate (Mauritzen, 1996; Mork and Skagseth, 2010; Chafik et al., 2015; Smedsrud et al., 2022; Almeida et al., 2023). In this study, the focus lies on observed changes of the cooling of the AW within the NwAC core as it propagates northward and the processes governing it.





The NwAC consists of two main branches: the Norwegian Atlantic Slope Current (NwASC) and the Norwegian Atlantic Front Current (NwAFC, figure 1a). The NwASC follows the continental slope northward, exhibiting a mostlybarotropic structure, while the NwAFC follows the Mohn Ridge in the central Nordic Seas (Orvik and Niiler, 2002; Skagseth et al., 2008; Mork and Skagseth, 2010). At the southern Norwegian location of Svinøy (∼62.8°N), the volume transport of the NwAC has been estimated at 5.1 ± 0.3 Sv, whereof about 3.4 ± 0.3 Sv are associated with the NwASC, which is the focus of this study

(Mork and Skagseth, 2010). There is a plethora of literature detailing the substantial spatio-temporal variability of NwASC transports along the continental slope (e.g. Mork and Skagseth, 2010; Chafik et al., 2015; Fer et al., 2020; Beszczynska-Möller et al., 2012) and the origin and propagation of thermohaline anomalies within (Furevik, 2001; Carton et al., 2011; Yashayaev and Seidov, 2015).

    The current follows the continental slope northwards and bifurkates north of the Norwegian mainland into a branch flowing

eastwards into the relatively shallow Barents Sea through the Barents Sea Opening (BSO, with a maximum depth of ∼450 m deep)(Loeng, 1991; Ingvaldsen et al., 2002) and another branch continuing nortwards along the continental slope, forming the West Spitsbergen Current (WSC) headed to Fram Strait (Beszczynska-Möller et al., 2012).

    Whether AW enters the BSO or continues towards Fram Strait is connected to the large scale atmospheric pattern (Lien et al., 2013; Heukamp et al., 2023). For example, wind-driven Ekman transports can cause negative sea-level anomalies around

Svalbard, which in turn effect anti-cyclonic current anomalies around the archipelago, yielding enhanced flow through BSO, while reducing transport through Fram Strait (Lien et al., 2013). Similarly, the pan-Arctic atmospheric Arctic Dipole oscillation has been identified as "switchgear mechanism" between BSO and Fram Strait on multi-annual time scales (Polyakov et al., 2023). On multiannual time scales, the inflow of AW into the Barents Sea has also been connected to the NAO and the Atlantic Multidecadal Oscillation, with regional hydrographic responses lagging the indices by approximately 4–5 years (Yashayaev

and Seidov, 2015).

    The climatic relevance for the NwASC stems not only from the transport of AW heat, but also from the substantial transformation (i.e. cooling and freshening) these waters undergo on their way towards the Arctic Ocean. For a detailed review and description of estimates of heat fluxes and (atmospheric) processes associated with the cooling, see Smedsrud et al. (2022) and the references therein.

The main driver for cooling is surface heat fluxes, as the ocean is warmer than the atmosphere for most of the year (Skagseth et al., 2020; Smedsrud et al., 2022). Over the last decades, air temperatures over the Nordic Seas have been rising faster (∼ 1°C decade$^{-1}$) than sea-surface (or skin) temperatures in the region (Isaksen et al., 2022; Mayer et al., 2023), thus reducing net surface heat fluxes from the ocean to the atmosphere.

    Mesoscale eddy activity also plays a critical role in lateral exchange and the heat budget of the NwASC. Eddy kinetic energy

and lateral diffusivity are highest near the Lofoten Escarpment (Andersson et al., 2011), where eddies facilitate cross-slope transport. Westward eddy heat flux accounts for roughly one-third of the NwASC's total heat loss (Bashmachnikov et al., 2023). In a study based on hydrographic profiles, combined with ocean reanalyses and idealized model considerations, Huang et al. (2023) claim that lateral heat transfer is the dominant mechanism for NwASC heat loss (as opposed to surface heat fluxes for the NwAFC).





Changes of the speed of advection of AW within the NwASC may impact the time the water mass is exposed to cooling through surface heat fluxes. The investigation into changes of velocities is non-trivial because transport rates of the NwASC vary substantially along the pathway (Chafik et al., 2015). The picture is further complicated by the finding that temperature anomalies may propagate poleward at much slower speeds (by up to a factor of 10) than the observed flow of the current (Årthun et al., 2017; Broomé and Nilsson, 2018). This difference is attributed to shear dispersion, in which anomalies are

mixed into slower-moving ambient waters, reducing their effective propagation speed (Broomé and Nilsson, 2018). Spiciness-based cross-correlation between Svinøy and the BSO supports this interpretation (Bosse et al., 2018). Furthermore, temperature anomalies tend to propagate more rapidly than salinity anomalies, indicating different controlling mechanisms—advection for salinity and air–sea exchange for temperature (Yashayaev and Seidov, 2015).

In this study, we use 30 years of direct ship-based measurements from four transects across the NwASC together with

satellite-altimetry based estimates of surface geostrophic velocities and surface heat fluxes from reanalysis products to investigate changes in cooling of AW within the NwASC core and the associated drivers. In particular we will consider three possible mechanisms impacting the cooling withing the NwASC core: 1. changes in surface heat fluxes, 2. changes in lateral heat fluxes and 3. changes in advection speeds.

## 2 Data & Methods

### 2.1 Hydrographic Data

The Institute of Marine Research (IMR) has been maintaining standard hydrographic CTD (Conductivity, Temperature and Depth) sections across the Norwegian continental slope and the Barents Sea Opening (BSO) that are revisited generally several (up to six) times a year for over 50 years as part of their Norwegian and Barents Seas monitoring program. Here we use data from the most frequented sections called in order from south to north: Svinøy (SI, 17 stations), Gimsøy (GI, 16 stations),

Barents Sea Opening (BSO, 20 stations) and Bjørnøya (BI, 12 stations, figure 1). We use data spanning the recent era with available satellite altimeter observations: 1993-2022. After automated quality control, which corrects or discards individual profiles and checks transects for completeness (detailed description in the appendix), we are left with a total of 412 transects performed over the last 30 years. However, the distribution of these sections is irregular in time (figure 2). For all sections, the timing is heavily seasonally skewed, with spring and summer (MAM & JJA) accounting together for 292 transects (70%). SI,

GI and BSO were frequented quite regularly over the years, amounting to 123, 105 and 131 repetitions, respectively, or average return times of 89 days, 102 days and 83 days. BI was only repeated 53 times over the 30 year period.

### 2.1.1 Data processing

With the hydrographic transects, we obtain a snap-shot of the water column about every 3 months. This snap-shot may contain (sub-) mesoscale variability in the hydrographic properties caused for example by eddies and tides (a transect typically takes

longer than one tidal cycle). For the purposes of this study, we assume that one transect should be representative of the ocean



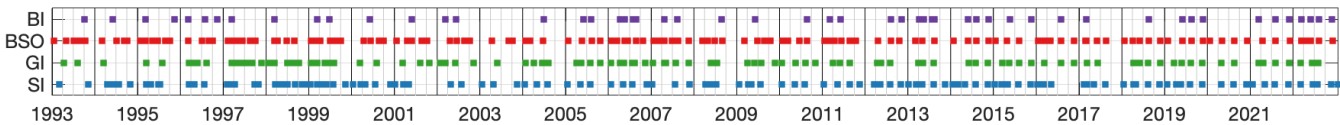

**Figure 1.** a) Overview map showing the Nordic Seas and the sections indicated as black lines (SI: Svinøy, GI: Gimsøy, BSO: Barents Sea Opening, BI: Bjørnøya). Selected isobaths are given as black contour lines and color shading is satellite altimetry based surface geostrophic speed averaged over 1993-2022. Labeled in the map are the two branches of the Norwegian Atlantic Current: the Norwegian Atlantic Front Current (NwAFC) and the Norwegian Atlantic Slope Current (NwASC). b)-m): Conservative temperature ($\Theta$), absolute salinity ($S_A$) and absolute geostrophic velocity (AGV) averaged over 1993-2022 for each section. Black contours are selected isopycnals and black triangles indicate locations of CTD profiles. For AGV (d,f,j,m), colored boxes at the surface indicate location and strength of average satellite altimetry based surface geostrophic velocities that are added to relative geostrophic velocity to create AGV sections.

**Figure 2.** Timing of each transect of the sections shown in figure 1.





state on seasonal time scales (i.e. about 3 months). Accordingly, we endeavor to remove (sub-) mesoscale variability from the transects. Gaussian smoothing of conservative temperature ($\Theta$, hereafter temperature) and absolute salinity ($S_A$, hereafter salinity) in horizontal and vertical direction often causes instability in the density stratification. Instead, we find it advantageous to effectively smooth the isopycnals in horizontal direction. This is achieved by interpolating salinity, temperature and the

associated field of depths onto density coordinates. Within the density coordinates, we smooth the depths along the horizontal direction with a running mean over three stations. Since CTD stations are not always at equal distances, the smoothing distance varies. In practice this leaves us with a higher effective resolution over the continental slope, where CTD stations are close and density gradients large and effective smoothing over several Rossby-radii further offshore. The fields of temperature and salinity are then re-interpolated from the smoothed depth field to the regular depth vector and density is recalculated. This

approach yields only minimal instabilities which are resolved by sorting the vertical profiles of temperature, salinity and density to ensure stability. From the smoothed, stable fields of temperature and salinity, we calculate relative geostrophic velocities (RGV) relative to the surface using the Gibbs-Seawater toolbox for Matlab (McDougall and Barker, 2011). Since RGV is based on horizontal gradients, it is calculated between pairs of profiles. In order to avoid large gaps of data at the bottom over the steep topography of the slopes, we first extend temperature and salinity of the shallower profile downwards until it reaches the

depth of the deeper profile. For this, we linearly interpolate between the deepest values of the shallower profile and the deepest values of the deeper profile, so that at maximum depth the horizontal gradient is zero. If needed, the extended profile was once again re-ordered by density to maintain stable stratification (in practice only few minor inversions occurred).

Surface geostrophic velocities from satellite-based altimetry measurements are available since 1993 on a $0.125° \times 0.125°$ grid on daily resolution (Global Ocean Gridded L 4 Sea Surface Heights And Derived Variables Reprocessed 1993 Ongoing: https:

//doi.org/10.48670/moi-00148). To match the presumed seasonal resolution of the hydrographic transects, the satellite-derived values for surface geostrophic velocities are smoothed by running mean over a 90-day (3-month) period before interpolating them to the locations of RGV data points (in-between original CTD stations). Adding the satellite-derived surface geostrophic velocities to the field of RGV yields a field of absolute geostrophic velocity (AGV) for each transect.

### 2.1.2 Defining the AW current core and creating time series

Obtaining meaningful and comparable time series of properties from different sections of varying spatial extent can be non-trivial. In this study, we focus on the properties of the NwASC core, which we can compare between sections. We first bound the area for each section based on 30-year mean properties as follows:

- Contains Atlantic Water (AW), defined via $S_A \geq 35.16$ g kg$^{-1}$, following the definition by Mork and Skagseth (2010); Ingvaldsen (2005); Loeng (1991)

- Has an upper boundary of 50 m below sea surface (to exclude large parts of the surface mixed layer)

- Is located shore-ward of the 1500 m isobath (SI) or 2500 m isobath (GI), to avoid the NwAFC and large scale recirculation, respectively.



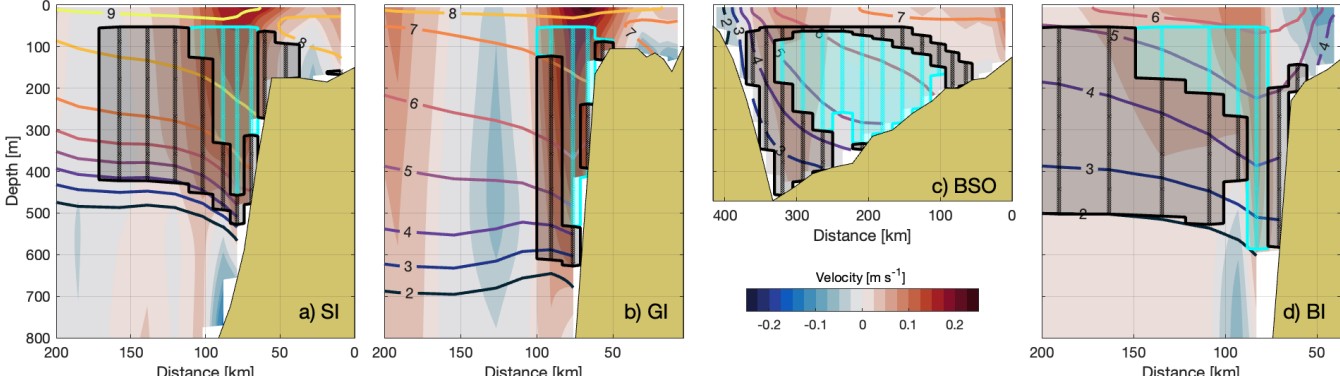

**Figure 3.** Sections of average AGV (as figure 1 d,g,j,m, but zoomed in on the continental slope), with selected isotherms as colored contours. Areas (shading) and actual data points (markers) comprising the core and non-core as defined in Sect. 2.1.2 are shown in cyan and gray, respectively.

Each data point within these areas is associated with a transport, calculated by multiplying AGV at the respective point with the area it represents. Cumulative summation of the transports within the area yields total AW transport through the section. At

Svinøy, Gimsøy and Bjørnøya, the NwASC core is defined to consist of the data points - sorted by decreasing AGV - needed to account for 50% of the total mean transport through the respective section. This yields an NwASC core defined by the strongest currents within the AW layer (figure 3 a,b,d). At BSO, the situation is different; the topography is a lot shallower and there is no pronounced mean boundary current (figure 1 g). Thus, the strongest mean currents do not represent the core of the AW. Instead, we use salinity so that the core is defined to consist of the data points - sorted by decreasing salinity - needed

to account for 50% of the maximum mean transport through the section. Analogous to above, this yields a core consisting the highest salinity waters as the NwASC is the sole source for high salinity waters in the region (figure 3 c). Core (and non-core) properties are then averaged within the respective core (or non-core) area for each transect at each section. This core definition is based on average properties and thus static over time. Experiments with dynamic definition of core area for each individual transect yielded some variability, but no substantial (nor significant) trends in area or location of the core (not shown). We

further note that for the scope of this study, the properties of the core are qualitatively insensitive to the choice of percentage of max transport (tested in a range from 50-70%).

To make the time series of the individual sections comparable, we interpolate them onto a common time vector with 3-month time resolution. While the time resolution corresponds roughly to the average return time for the SI, GI and BSO sections, this does not take into account the seasonal bias of the actual transects (figure 2). For BI, the 3 month interpolation corresponds

on average to an oversampling and care has to be taken interpreting results based on this interpolation. For some analyses, de-seasoning is necessary, which is performed by 12-month moving averaging of the 3-month interpolated time series.





### 2.1.3  Correlations

Due to the substantial seasonality of some properties (e.g. temperature), correlations are done using the annually smoothed and de-trended 3-month interpolated time series (except when stated otherwise). A side effect of the smoothing is that neighboring

points are not independent from each-other which has consequences for the calculation of the correlation coefficient. To account for the auto-regression, significance intervals for all correlation coefficients presented in this study are calculated using the effective degrees of freedom via the modified Chelton method presented in Pyper and Peterman (1998).

### 2.2  Heat fluxes

Heat fluxes and other meteorological parameters (2 m temperature, 10 m wind and sea-surface temperature) in the study domain

are obtained from ERA5 reanalysis on a $0.25° \times 0.25°$ grid and provided as monthly averages (Hersbach et al., 2020). We define positive heat fluxes to be directed upwards from the ocean to the atmosphere.

### 2.3  Box model estimates

The impact surface heat fluxes have on ocean temperatures, can be estimated using a simple box model. In particular, we solve the following equation

$$\Delta T = H_f \ B_{surf} \ t/(\rho_w \ B_{vol} \ cp_w) \tag{1}$$

with the temperature change $\Delta T$ equaling the surface heat fluxes $H_f$ (obtained from ERA5), acting over the duration of the (advection) time $t$ on the surface of the box $B_{surf}$, impacting a volume of water defined by the density of sea water $\rho_w = 1027.6$ kg m$^{-3}$ (corresponding to the average NwASC core water density at Gimsøy), the volume of the box $B_{vol}$, and the specific heat of water $cp_w = 3.9919 \times 10^3$ J kg$^{-1}$ K$^{-1}$. The dimension of the box is given by the respective length of the pathway

between two sections (about 750 km for the path between Gimsøy and Bjørnøya), the width of the box set to 50 km (based of figure 3 and Huang et al. (2023)). This yields a surface of the box $B_{surf} = 750 \times 10^3 \ m \times 50 \times 10^3 \ m = 3.75 \times 10^{10} \ m^2$. The area of the cross-section is set to 10 km$^2$, which is representative of the core areas at Svinøy, Gimsøy and Bjørnøya (about 11 km$^2$, 8 km$^2$ and 17 km$^2$, respectively). This yields a depth of the box of 200 m, with the total volume of the box $B_{vol} = B_{surf} \times 200 \ m = 7.5 \times 10^{12} \ m^3$. Due to the absence of any vertical mixing parameterization in the box model, absolute

values obtained for $\Delta T$ will be heavily biased, but anomalies will be interpretable under the assumption of approximately linear scaling of mixing with respect to surface heat fluxes.

## 3  Results

### 3.1  Time-average properties of the NwASC

The spatial extent, as well as the shape of the core and non-core varies substantially between sections (figure 3). At Svinøy,

the core is wedge-shaped, centered roughly above the 600 m isobath and reaching a maximum depth of 460 m (figure 3a). At





50 m it has a lateral extent of about 50 km, comprising 4 CTD casts. Horizontally, the non-core area is limited by the 1500 m isobath, which is about 150 km offshore and vertically reaching down to 530 m where it is bound by salinity. At Gimsøy, the exceedingly steep topography yields a vertically more stretched core and non-core, reaching down to 530 m and 630 m respectively while each only extending about 50 km horizontally due to the defined boundary at 2500 m isobath (figure 3b). In the Barents Sea, where the core is defined via salinity, it makes up large parts of the water column over the shallow southern slope down to about 350 m water depth (figure 3c). The non-core area surrounds the core, with the largest area located over the deep trench and the steep northern slope of the BSO. The core and non-core areas at Bjørnøya strongly resemble those at Svinøy, albeit reaching slightly deeper (about 580 m, figure 3d).

In a time-mean sense, the $\Theta$-S$_A$ properties of both core and non-core areas show a systematic cooling and freshening from south to north (figure 4). The core always contains warmer and more saline waters than the non-core, which may be readily explained by lateral mixing with surrounding water masses and increased vertical heat fluxes due to the slower, possibly more meandrous propagation of water in the non-core areas. In the BSO, the core is by definition more saline than the non-core, but it is also substantially warmer than the BI core. This may be partly due to a somewhat shorter pathway from GI to BSO compared to GI to BI, and partly due to extensive mixing between the water in southern part of BSO core and non-core areas and the warm and fresh coastal current, whose $\Theta$-S$_A$ properties show clear mixing-lines with the core and non-core $\Theta$-S$_A$ properties (not shown).

Average transport estimates are by definition identical for core and non-core areas of each section. The combined core and non-core transport is 2.60 Sv, 2.29 Sv, 1.51 Sv and 2.17 Sv for the Svinøy, Gimsøy, BSO and Bjørnøya sections, respectively. This is within the range of values presented in the literature for various periods (Mork and Skagseth, 2010; Fer et al., 2020; Ingvaldsen et al., 2004; Beszczynska-Möller et al., 2012).

### 3.2 Temporal variability of NwASC properties

Time series of temperature, salinity and transport exhibit substantial inter-annual variability throughout the 30-year time period (figure 5). Especially notable are the significant trends of temperature at all sections for both core and non-core area averages (bold font in the legend, figure 5a-d). For salinity, there are no significant trends anywhere, but the multi-annual variability is most pronounced with an increase of over 0.1 g kg$^{-1}$ during 1995-2010 at all sections, followed by a near-synchronous more rapid decrease after 2015 (figure 5e-h). Interestingly, this change of salinity does not appear to have substantially impacted density, which exhibits variability and significant negative trends in accordance with changes in temperature (not shown). While showing substantial inter-annual variability, there does not appear to be any structured multi-annual variability in RGV (not shown) or transports, nor are there any significant trends (figure 5i-l).

### 3.3 Long-term reduction in cooling of NwASC north of Gimsøy and its potential drivers

In a steady regime of cooling and freshening, the difference of temperature and salinity between sections may be constant regardless of overall trends in the properties. Between Svinøy and Gimsøy, variability of temperature differences is about 1 °C and the trend of -0.03 °C decade$^{-1}$ is not significantly different from zero at the 95% confidence interval (table 1).





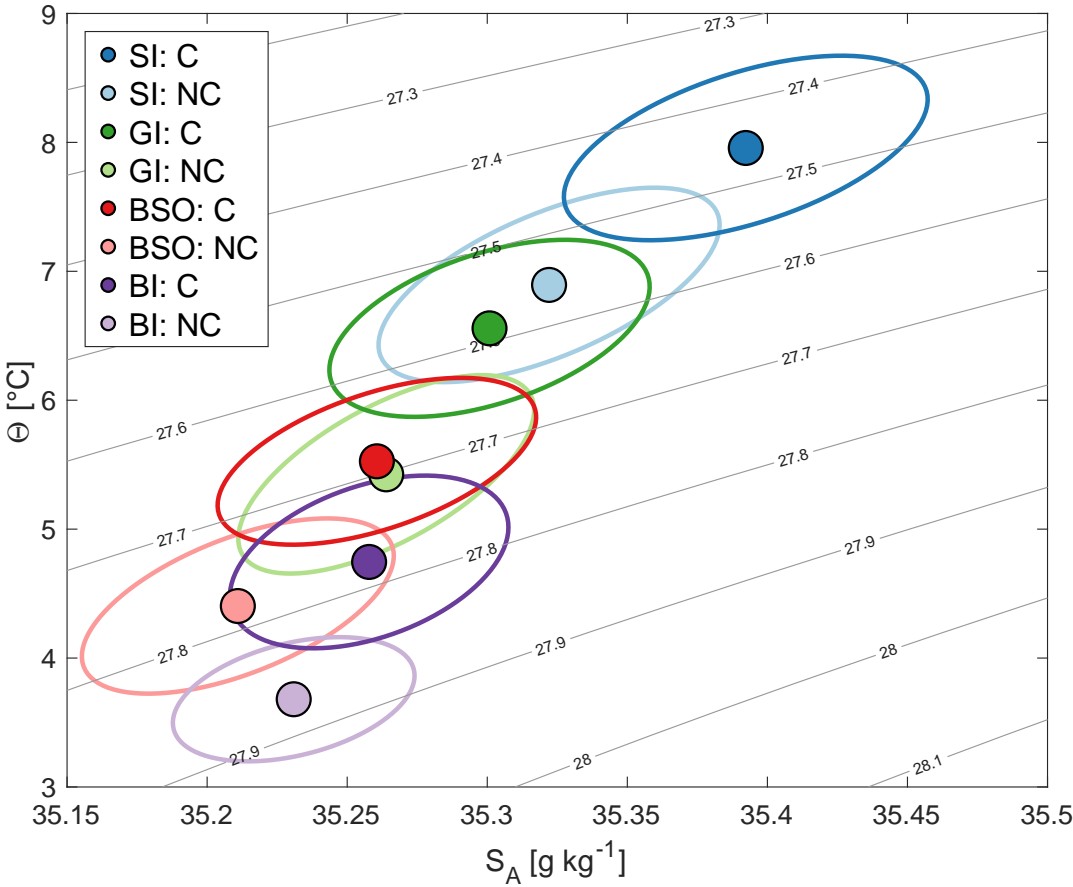

**Figure 4.** $\Theta$-$S_A$ diagram of time-average properties averaged within the core (C) and non-core (NC) of each section. Ellipses indicate 50% variance.

Going further north, differences between both Svinøy and BSO as well as between Svinøy and Bjørnøya exhibit significant decreasing trends of -0.13 °C decade$^{-1}$ each. The absence of a trend between Svinøy and Gimsøy suggests that most of these trends manifest north of Gimsøy. Indeed, trends in the temperature difference between Gimsøy and BSO as well as Gimsøy and Bjørnøya are almost identical to those relative to Svinøy, with -0.11 °C decade$^{-1}$ and -0.13 °C decade$^{-1}$, respectively (table 1). There is no significant trend in temperature difference between BSO and Bjørnøya (table 1). Trends of salinity differences are also negative throughout, but only significant for all differences involving Bjørnøya. Amplitudes are small ($\leq$ 0.016 g kg$^{-1}$ decade$^{-1}$, table 1). In summary, these results indicate reduced cooling of AW between Gimsøy and BSO/Bjørnøya and thus point towards a change of heat loss mechanisms.





**Figure 5.** Time series of core (solid colors) and non-core (lighter colors) properties for each section, both in original time resolution (thin lines) and on annually smoothed 3-month interpolated time vector (thick lines). Legends in each panel show the linear trend (calculated from the original, un-interpolated time series) and correlation to the core time series. Bold font in the legend indicates trends that are different from zero at 95% confidence.

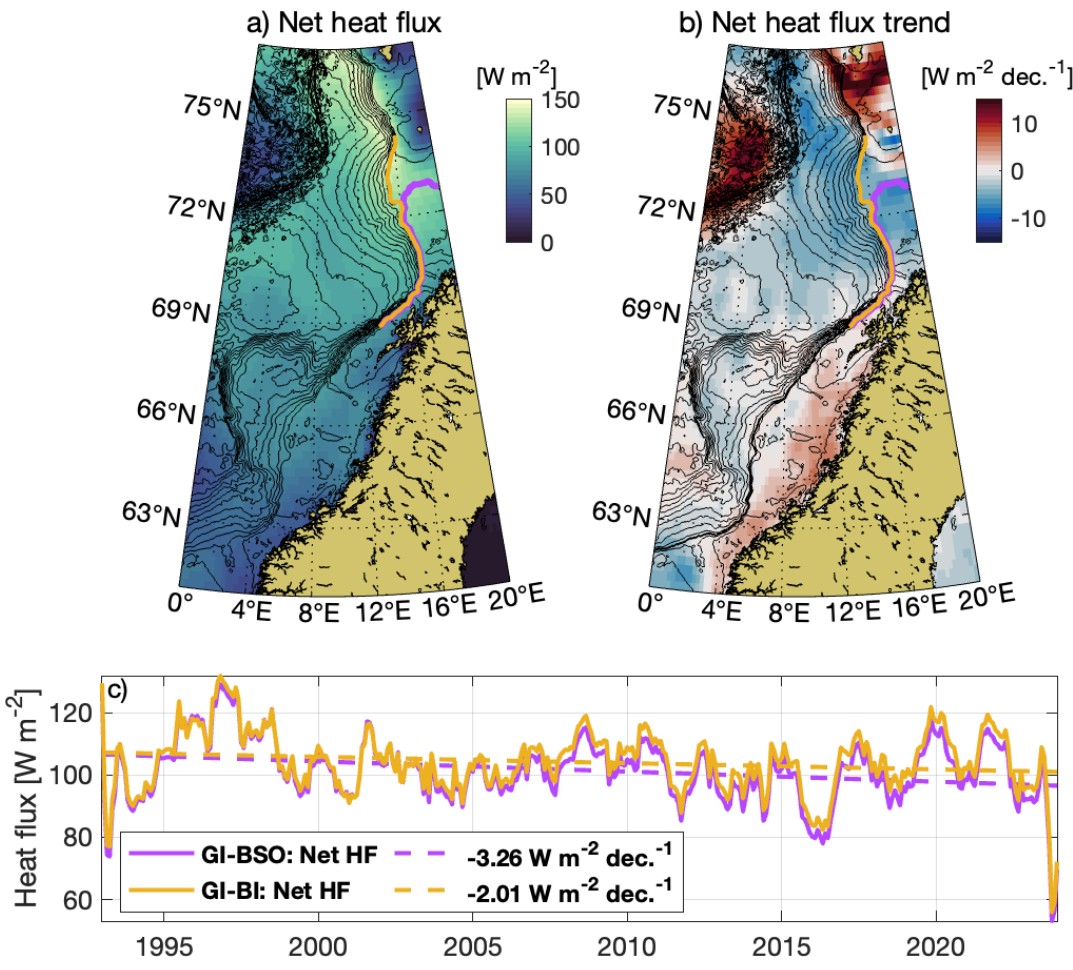

**Figure 6.** a) Net heat flux averaged over 1993-2022. b) Linear trend of net heat flux. c) Time series and linear trends of the net heat flux averaged over the line segment along the 400 m isobath between GI and BSO (purple) and for the line segment along the 500 m isobath between GI and BI (yellow). As before, significant linear trends are written in bold font.





**Table 1.** Trends of temperature differences calculated pairwise between all sections for core temperatures. As before, bold font indicates trends that are significantly different from 0 at 95% confidence.

| Section Pair | $\Delta\Theta$ core trend [°C decade$^{-1}$] | $\Delta S_A$ core trend [g kg$^{-1}$ decade$^{-1}$] |
|---|---|---|
| SI - GI | -0.03 | -0.004 |
| SI - BSO | **-0.14** | -0.007 |
| SI - BI | **-0.14** | **-0.016** |
| GI - BSO | **-0.11** | -0.004 |
| GI - BI | **-0.13** | **-0.011** |
| BSO - BI | 0.00 | **-0.006** |

### 3.3.1 Reduced cooling through the surface

Surface heat fluxes account for the largest loss of heat from the ocean in the Nordic Seas, with 30-year averages reaching 60 W m$^{-2}$ over the Nordic Seas (full domain of figure figure 6a), and exceeding 100 W m$^{-2}$ in the vicinity of BSO and Bjørnøya (figure 6a). Over the last 30 years, surface heat fluxes have decreased substantially, specifically in the region between Gimsøy and BSO/Bjørnøya (figure 6b). In particular along the pathway of the AW entering the BSO following the 400 m isobath, net heat flux from the ocean to the atmosphere has reduced by 3.26 W m$^{-2}$ decade$^{-1}$, or about 3% per decade (figure 6c). Notably all components contributing to net heat flux (sensible heat flux, latent heat flux, shortwave radiation and long wave radiation) show significant trends with shortwave radiation being the only component with a positive trend (0.45 W m$^{-2}$ decade$^{-1}$) and sensible heat flux accounting for the vast majority of the net decrease with -2.58 W m$^{-2}$ decade$^{-1}$ (not shown). When following the pathway of NwASC along the 500 m isobath between Gimsøy and Bjørnøya, the negative net heat loss trend is somewhat smaller with -2.01 W m$^{-2}$ decade$^{-1}$ and sensible heat flux is the only component with a significant trend (-2.15 W m$^{-2}$ decade$^{-1}$). The reduced net heat flux from the ocean to the atmosphere causes less cooling and may thus contribute to the decrease in core temperature difference between GI and BSO/Bjørnøya (table 1). The decrease in surface heat fluxes is mainly due to a decline in sensible heat flux.

### 3.3.2 Reduced cooling by reduced lateral heat flux

It has been estimated that eddy-driven lateral heat transport accounts for about one third of heat loss from the NwASC (Bashmachnikov et al., 2023). However, lateral heat fluxes (e.g. via baroclinic and/or barotropic energy conversion) are difficult to directly observe and require dedicated observational efforts. The only long term moored observations in the region are located at the Svinøy section over the 500 m isobath. There, moored current meters have been operational continuously over the whole period of interest in this study (1993-2022). EKE is calculated at 100 m water depth using daily bin-averaged velocity fluctuations as deviations from 30-day averaged means and then interpolated on the common 3-month time vector. The average EKE is 0.0087 m$^2$s$^{-2}$ and shows a slightly negative linear trend over 30 years (-6.4327×10$^{-4}$ m$^2$s$^{-2}$decade$^{-1}$) that is not signifi-



cant (p=0.102). Decreased (or increased) lateral heat fluxes, could be expected to increase (or decrease) the lateral temperature
gradient. This is not observed at any section crossing the NwASC. The verdict on whether the lateral heat flux has changed in
the area thus remains inconclusive.

### 3.3.3 Increased advection speed leaves less time to cool

Time series of RGV and transports do not show any significant trends at any section (figure 5). However these are only spatial
snap-shots and do not capture possible changes along the pathway. Surface geostrophic currents along the whole pathway of the
NwASC are obtained from satellite altimetry (the same product used to calculate AGV from RGV, see Sect. 2.1.1). Following
the 500 m isobath between Svinøya and Gimsøy, along-topography currents average 0.17 m s$^{-1}$ with a small, insignificant
trend of -0.0021 m s$^{-1}$decade$^{-1}$. Between Gimsøy and BSO, following the 400 m isobath (shown in figure 6a), the average
along-topography current is reduced to 0.12 m s$^{-1}$ with an insignificant trend of -0.001 m s$^{-1}$decade$^{-1}$. On the segment
between Gimsøy and Bjørnøya (figure 6a), the trend is even smaller at -0.0005 m s$^{-1}$decade$^{-1}$. While all of these trends are
negative, they are minute and none of them is significant at 95% confidence (when smoothed and interpolated on the 3-months
common time vector).

An independent measure of advection velocity along the NwASC lies in the tracing of anomalies as they propagate through
the sections. Because of its small influence on density in the domain, salinity acts largely as a tracer and is thus best suited to
investigate advection speed of signals, and shifts in these, along the continental slope. However, estimating cross-correlations
and associated lags of these unevenly sampled time series proved non-trivial. We use 12-month smoothed salinity time series
and calculate cross-correlations in various windows (ranging from 5 to 15 years in size), sliding at at increments of 1/4 window
length. As the window slides over the time series, the lag of the highest cross-correlation within each window is calculated.
Only results with satisfactory data coverage (at least 50% of expected quarterly data points within any window), significant
correlation coefficients (at 95% confidence) and sensible lags (between 0 and 24 months) were taken into account. Due to
sparse sampling, Bjørnøya was excluded from this analysis. Lags are plotted centered within their respective window (colored
dots) and two-year bin-averaged lags are shown as black lines in figure 7. While the effective smoothing is uneven and ranges
between 5 and 15 years, depending on the respective window, tests showed good temporal fidelity of the analysis. Reassuringly,
the sum of the mean lags calculated in the segments Svinøy to Gimsøy and Gimsøy to BSO, figure 7a,b) corresponds well with
the estimate of lags between Svinøy and BSO (i.e. they agree within the 3-month temporal resolution, not shown). The estimates
show substantial variability of advection times, ranging between 0 and 12 months. Despite an absence of reliable results before
2000, there is a clear and robust pattern visible in both segments. In particular, on the segment between Svinøy and Gimsøy
(figure 7a), lags of 0-3 months are observed until ∼2010, followed by a marked increase of lags to about 9 months around
2015-2019. The final available data points indicate again a decrease in lags. On the segment Gimsøy and BSO (figure 7b), lags
of 9-12 months dominate the time interval from 2002 to 2011. After that there is a continuous and robust reduction in lags
to 0-3 months. Such drastic variability is not represented in either transports (5i-l), nor satellite-derived surface geostrophic
currents (not shown). However, it is known that advection speed of anomalies can differ substantially from observed current
speeds due to internal processes such as velocity shear (Broomé and Nilsson, 2018). Indeed, time series of total vertical shear





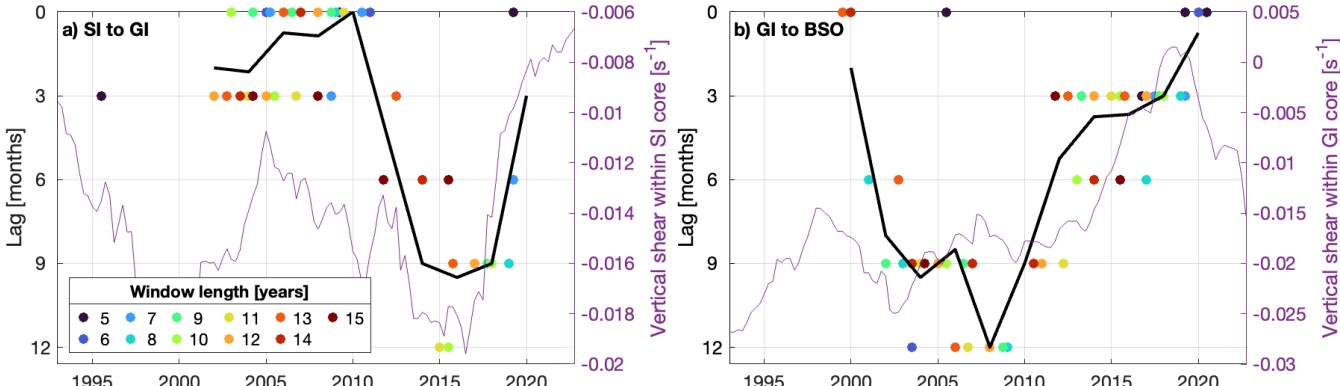

**Figure 7.** Lags of maximum cross-correlation of 12-month smoothed $S_A$ between SI and GI (a) and GI and BSO (b) within various windows ranging from 5-15 years (colors of dots), sliding at increments of $1/4$ window length. Shown are only results with significant correlation coefficients ($p \leq 0.05$), positive lags (between 0 and 24 months), and where both time series have at least 50% of the expected (quarterly) data points in any given window (this is why the less frequently sampled BI section is not represented here). The black line represents bin-averaged lags in two-year bins. The purple line shows time series of total vertical shear within the core at Svinøy (a) and Gimsøy (b), smoothed with a 5-year running mean.

estimated from the AGV within the cores at the Svinøy and Gimsøy sections show variability similar to the cross-correlation analysis (figure 7a,b purple lines), with decreasing lags in recent years on the Gimsøy - BSO segment coinciding with reduced

vertical shear observed at Gimsøy (note that total vertical shear is generally negative here, such that a reduction manifests in a rising curve).

## 3.4    Variability of NwASC cooling north of Gimsøy

Inter-annual variability of cooling within the NwASC north of Gimsøy can be estimated by calculating the anomalous temperature difference ($\Delta\Theta$) between Gimsøy and BSO (Bjørnøya is again excluded from this analysis due to sparse sampling) using

plausible advection times of 3-12 months (figure 7) to capture the range associated with advection speed uncertainty. To emphasize longer term variability, a smoothing of 3 years is applied (figure 8, blue line and shading). The largest $\Delta\Theta$ anomalies are observed around 1997 and 2003. After 2005, amplitudes of $\Delta\Theta$ anomalies decline and become more negative, reflecting the overall reduction of cooling shown in table 1. The most negative $\Delta\Theta$ anomalies are found between 2017 and 2020 and appear to be mostly due to two individual strong negative temperature anomalies at Gimsøy, that are not represented at any other

section (figure 5a-d). To estimate how far the observed variability of cooling may be attributed to variability of surface heat fluxes, a simple box model is used (Sect. 2.3). The box model calculates the change in temperature of a volume of water due to realistic 3-year-smoothed ERA5 heat fluxes for a given advection time. It is important to note, that in the absence of a realistic mixing parameterization, absolute values of the box model results will be biased and results are only meaningful as anomalies. Since advection time act as scaling factor in the model (equation 1), the anomalies calculated using different advection times (3,





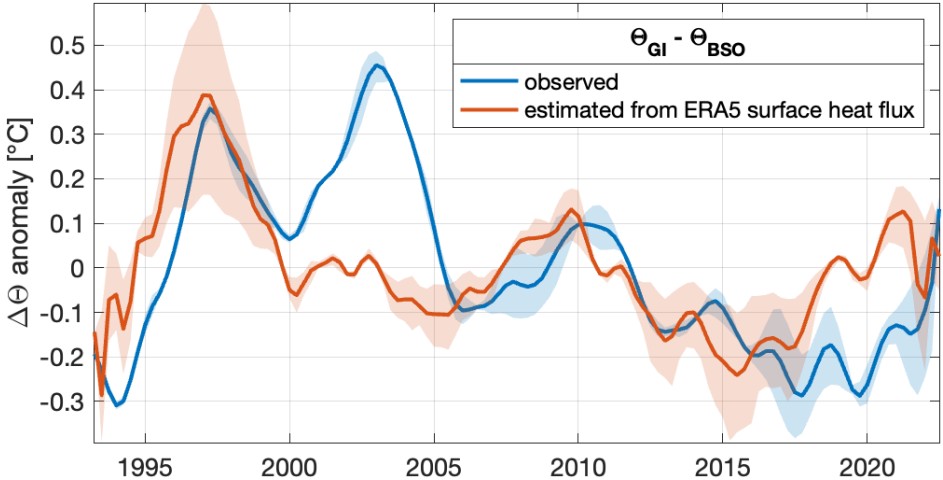

**Figure 8.** 3-year smoothed time series of $\Delta\Theta$ core anomalies between Gimsøy and BSO. Blue are observations, red are estimated from ERA5 heat fluxes using a simple box model. Shading is the envelope of results using different advection times, ranging between 3 and 12 months (cf. figure 7b), solid lines are the average of the envelope.

6, 9 and 12 months) all have different base-lines. Box model estimates closely follow the observed $\Delta\Theta$ anomaly peak between 1995 and 2000, as well as variability from 2005 to 2017 (figure 8, red line and shading). This suggests that surface heat flux variability can explain much of the cooling variability in those periods. Larger discrepancies arise during 2000–2005, when observed anomalies exceed model estimates by $\sim$0.4°C, and during 2017–2021 when model estimates exceed observations by $\sim$0.2°C. In these periods, surface heat loss changes alone cannot account for the observed cooling variability.

**4   Discussion**

This study presents robust observational evidence of a long-term reduction in the cooling of AW in the NwASC core north of Gimsøy, particularly between the Gimsøy and the BSO and Bjørnøya sections. The trend in core temperature differences between these sections exceeds -0.11°C decade$^{-1}$ (table 1), with the strongest signal observed in the Gimsøy–Bjørnøya segment. Our results indicate that both reduced surface heat fluxes and faster advection speeds likely contribute to the observed trends,
whereas lateral heat loss does not appear to have undergone any systematic change. Simple box model calculations further point towards time-varying interplay between drivers determining heat loss on the NwASC path north. Below, we discuss our key results in the context of existing literature.

**4.1   Reduced surface heat loss as a primary driver of change**

Net surface heat loss along the NwASC pathway between Gimsøy and BSO/Bjørnøya has significantly declined by $\sim$3%
per decade over the past three decades, with trends exceeding -2 W m$^{-2}$ decade$^{-1}$ along the 400 m isobath (figure 6). This





reduction stems mostly from the northeastern part of the Lofoten Basin and is dominated by decreasing sensible heat fluxes. Drivers behind this trend are reduced air-sea temperature gradients due to the former warming faster than the latter, which Mayer et al. (2023) suggest may be attribute to more southeasterly winds, ultimately related to a strengthening of the Icelandic low that advect warmer air masses from lower latitudes. Simple box-model budget calculations (Sect. 2.3) indicate, that in

order to increase the temperature of a volume of water set to represent the NwASC core between Gimsøy and Bjørnøya by 0.13 °C, corresponding to the observed reduced cooling trend in table 1, heat fluxes of about 3.3 W m$^{-2}$ suffice, assuming an incubation time (i.e. advection time) of 12 months. While simplistic, these estimates suggest that surface heat flux may be the dominant driver for changes in the cooling along the NwASC. Surface heat fluxes are known to be a key driver of heat content variability in the region (e.g. Mork et al., 2019), but timescales and amplitudes of associated anomalies are under debate (Carton

et al., 2011; Yashayaev and Seidov, 2015). Furthermore, heat fluxes can be seen as both cause and consequence of a changing ocean heat: if the ocean warms relative to the atmosphere, heat fluxes increase, indicating increased cooling (Asbjørnsen et al., 2020). In terms of variability, our box model analysis using ERA5 heat fluxes indicate that surface fluxes reproduce observed temperature difference anomalies between Gimsøy and BSO for several time periods (Figure 8), particularly from 1995–2000 and 2005–2015. At the same time, discrepancies between modeled and observed anomalies (e.g., 2000–2005 and post-2017)

point towards other drivers of variability. Using the box model, we can estimate the relative changes in advection time that would be needed to account for the observed variability in temperature difference anomalies. In particular, in order to achieve the ~0.45°C anomaly observed in 2003 with the realistic ERA5 surface heat fluxes, advection speeds need to reduce by about 50% relative to the mean advection speed. Conversely, to account for the ~-0.3°C anomaly observed in 2017, an acceleration of ~30% relative to the mean advection speed is required. This is in qualitative agreement with the changes in advection

speed we deduce from the advection of salinity anomalies for the Gimsøy-BSO segment (Figure 7b). However, other processes such as increased baroclinic conversion at the Lofoten Escarpment (Fer et al., 2020) or general periods of interruptions and discontinuities (Chafik et al., 2015) may also be the reason that anomalous upstream signals at Gimsøy are not advected further north (see figure 5a-d).

## 4.2 Increased advection speed

While transports derived from geostrophic velocities from hydrographic transects and satellite-derived surface currents do not display significant long-term trends (figure 5i-l), independent evidence comes from salinity anomaly propagation. Cross-correlation analysis (figure7) reveals a substantial decrease in lag times between Gimsøy and BSO, where the average lag decreases from approximately 9-12 months in the early 2000s to 3-6 months after 2015. Although salinity anomalies do not necessarily travel at the same speed as temperature (or heat) anomalies (Yashayaev and Seidov, 2015), this implies an

acceleration of advection speed, reducing its residence time and thereby limiting heat loss to the atmosphere. The notion that advection speed changes while geostrophic transports (figure 5i-l) and satellite-derived surface currents (not shown) are steady may appear to be inconsistent, but it is well known that advection speeds can be up to 10 times lower than observed current speeds in the NwASC due to various processes including shear and lateral exchange (Årthun et al., 2017; Broomé and Nilsson, 2018). Indeed, there appears to be a co-variability between salinity advection speed and total vertical shear within the core at



Svinøy and Gimsøy (figure7, purple lines): reduced lag times generally coincide with times of reduced vertical shear and vice versa, indicating that a more barotropic current advects faster than a more baroclinic current. Furthermore, there is a long term reducing trend in total vertical shear at Gimsøy, corroborating the notion of increased advection speeds on the Gimsøy-BSO segment. While the sparse and noisy sampling of the lag time series makes quantitative comparisons challenging, the qualitative agreement with total vertical shear supports the robustness of the general results of the analysis. To find the drivers behind a change towards a more barotropic current at Gimsøy will require dedicated research, possibly starting with an investigation into changes of storminess and related rapid geostrophic adjustment in the area (Brown et al., 2023). Within the scope of this article, we argue that an increased speed of advection contributes to reduced heat loss north of Gimsøy.

### 4.3 No evidence for reduced lateral heat loss

Eddy-mediated lateral heat loss is a key component of the NwASC heat budget, with recent estimates suggesting it accounts for up to one-third of total heat loss (Brown et al., 2023; Bashmachnikov et al., 2023). However, our analysis finds no consistent evidence of a long-term change of lateral heat exchange that could explain the observed reduction in cooling. Specifically: EKE at 100 m depth from the long-term Svinøy mooring shows a slightly negative but statistically insignificant trend (Sect. 3.3.2). Horizontal temperature gradients along each section can be seen as a heuristic proxi of lateral exchange and also do not show any significant trends. There is a positive trend in satellite altimetry-based estimates of surface EKE which could indicate an increase in lateral exchange and thus increased cooling, but the trend appears to be spurious and not supported by any other measure. We note that our results contrast with the hypothesis by Huang et al. (2023) that lateral processes dominate cooling of AW in the NwASC. While lateral heat loss remains important, our findings suggest its long-term variability is limited or at least not sufficient to explain the reduction in cooling observed here.

### 5 Conclusions

Using 30 years of ship-based CTD observations in conjunction with satellite-altimetry derived surface geostrophic currents, we show that the AW within the NwASC core has undergone a long-term reduction in cooling north of Gimsøy. This is the case for both branches of the NwASC: The one going into the Barents Sea via BSO, and further north along the continental slope through the Bjørnøya section. We identify a reduction in surface heat fluxes by $\sim$3% per decade in the region to be a dominant driver behind this change, in conjunction with accelerated advection speeds. The latter are not visible in transport estimates and satellite-derived surface geostrophic currents, but manifest in a reduction in lag times of salinity anomalies between Gimsøy and BSO from approximately 9-12 months in the early 2000s to 3-6 months after 2015. This is likely due to reduced vertical shear within the NwASC core resulting in a more barotropic flow. We find no evidence for systematic changes in lateral heat transport, as approximated by EKE estimates and lateral gradients of temperature and salinity. In terms of inter-annual variability of cooling within the NwASC, variability of surface heat fluxes may be the dominant (or even sole) driver for several periods particularly from 1995–2000 and 2005–2015, whereas other processes, such as changes in advection speed are necessary to explain observed variability on cooling during other periods such as 2000–2005 and post-2017.



*Author contributions.* TMB has conceived the study, analyzed the data and written the manuscript. ØS, RBI and KAM have contributed data, provided feedback on the methods and helped write the manuscript.

*Competing interests.* The authors declare no competing interests

*Acknowledgements.* TMB, ØS and RBI acknowledge funding by the European Union as part of the EPOC project (Explaining and Predicting the Ocean Conveyor; grant number: 101059547). Views and opinions expressed are however those of the author(s) only and do not necessarily reflect those of the European Union. Neither the European Union nor the granting authority can be held responsible for them. We acknowledge the use of artificial intelligence (AI) tools in helping to improve the flow of the text. Any changes suggested by the AI tool were adjusted and fact-checked by the authors.



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
