# Peer review of "Reduced Cooling in the Norwegian Atlantic Slope Current: Investigating mechanisms of change from 30 years of observations"

_EGUsphere, 2025_

## Author Comment (AC1)

We thank both reviewers for their constructive comments and suggestions, which helped us to improve this manuscript. Here we detail our responses (in blue) to each of the points raised by the reviewers (black). Line numbers given by us refer to the revised manuscript (not the tracked-changes version).

**Reviewer #1**

**Review of Baumann et al.**

This work gives valuable updates on transports and hydrographic variability, including long-term trends, at standard monitoring sections along the Norwegian slope. Focus is on reduced cooling at the northern part of this slope, which is most directly linked to declining air-sea heat fluxes. The manuscript is generally well written although too much detail is provided in places. Weaker/less robust aspects of this work (eddy fluxes, advection times) occupy too much space, and the apparent ignorance of the outer Norwegian Atlantic Front Current may be a severe limitation.

If the less robust parts are trimmed and the aforementioned severe limitation addressed (as suggested below), this could become a strong contribution to the literature.

We thank the reviewer for their insightful comments and feedback. We carefully assessed suggested changes and implemented many of them. In some cases, our opinions differ and we provide our reasoning for not implementing a specific suggestion.

**Surface heat flux analysis**

Throughout this manuscript, it is stated that the reduced cooling is due to reduced surface heat fluxes (especially sensible heat fluxes). This is introduced as early as line 50 (although disputed already in line 58). I suggest that this main result should be better emphasized, while that the weaker components (time lag and lateral eddy fluxes) should be downplayed.

I would structure the Results as follows:

Section 3.1 is well-executed, and should remain first. This section is, however, somewhat lengthy; some details could/should be moved to the Data & Methods part. The associated figure 3 and 4 are appropriate.

We thank the reviewer for this suggestion. We moved the description of the core shapes to the method section, shortening 3.1 considerably.

Section 3.2 makes an important contribution.

Section 3.3 discusses trends in temperature differences between the standard sections, relying only on references to Table 1, with no figure. The temporal variability in the temperature differences between the Gimsøy Section (GI) and the Barents Sea Opening (BSO) is not substantively presented in Section 3.4 and Figure 8.

*Suggestion*: Integrate the information in Section 3.4 into 3.3, so that Table 1 and Figure 8 provide mutual support.

We followed this suggestion and found that it indeed improved the flow of the manuscript; thank you!

Section 3.3.1 describes the most plausible driver; therefore give this sufficient emphasis.

We changed the last sentence of the first paragraph to: "These estimates indicate that surface heat flux may be the dominant driver for changes in the cooling along the NwASC."

Section 3.3.2 presents a rather heuristic attempt to discuss eddy fluxes. This is based on current meter moorings at the Svinøy Section, which is far upstream from the main signal (the reduced lateral cooling north of Gimsøy).

*Suggestion:* Move this section to your Discussion rather than presenting it as a primary result.

We carefully considered this suggestion, but we think there is value in having all three plausible drivers presented together. Moreover, since we do original data analysis in this section, we believe the place in the results section is justified even though the results are rather inconclusive. We did shorten this section a little.

Section 3.3.3 is excessively long and detailed, without providing commensurate value to the paper.

Suggestion: Consider retaining this section in abbreviated form, showing only Figure 7b for the GI–BSO region, where your main signal is identified (e.g. Figure 8). The connections between the advective time lags and the discrepancy between the GI-BSO temperature difference and sea-to-air heat fluxes are not particularly strong. However, it is worth showing that the lags were short immediately after 2000 and after 2017, when the aforementioned discrepancy was large.

We thank the reviewer for pointing this out and we agree that this section was too long. We moved the description of the cross-correlation analysis into the method section and constrained the analysis to the GI-BSO section as suggested.

**Inner and outer current branches**

It is somewhat surprising that this work does not refer to Blindheim et al. (2000), whose abstract, amongst other findings, states: "A temperature rise in the narrowing Norwegian Atlantic Current is strongest in the north."

This 'narrowing' refers to the outer branch (NwAFC), which is not addressed in the present work. This represents a potentially serious limitation. The spatial windows for estimating time series at the GI and SI sections represent a predominantly barotropic core/slope current. This is, however, not the case for the BSO, where the window is much broader, likely including influence from the outer branch, and represents much less of a (likely swift) slope current. Parts of the NwAFC passing through the Svinøy section, will turn eastwards with the topography of the Vøring Plateau, feeding the outer branch at GI; and likely also contributing to parts of the area selected for the BSO. The reported advection time lags of up to 12 months between the relatively closely positioned GI and BSO likely involve this outer current branch, rather than solely a fast nearly barotropic slope current, as portrayed here. This alternative (and in my opinion more realistic) perspective including the outer branch would require adjustments to your box model. Most importantly would the surface area of the Atlantic waters (Bsurf) become variable, and this variability could be as important as changes in the air-sea heat fluxes (W m-2). Changes in advective time lags are also linked to the width of the boundary current system, and thus to Bsurf. The boundary current was weak/broad in 1997 and 2003, coinciding with when you observe that largest temperature differences between GI and the BSO (Figure 8).

Suggestion: Revise your description to incorporate the above-described, more realistic scenario, including appropriate reference to Blindheim et al (2000).

We thank the reviewer for bringing the issue of possible interaction between NwAFC and NwASC to our attention.

In the time-mean Eulerian sense, the NwASC and NwAFC are well-defined and separated from south of Svinøy all the way to Fram Strait (e.g. figure 1 in the manuscript or Huang et al. 2023, their figure 1b). However, recent studies employing Lagrangian trajectories of both observed and simulated tracers suggest the potential for exchange between the NwASC and NwAFC, particularly around the eddy-rich Lofoten Basin (Ypma et al., 2020, Broomé et al., 2021).

While there clearly is evidence for *some* exchange between the currents, it is unclear wheter this amounts to a significant transport of water from the NwAFC and NwASC/BSO (interestingly, in the study by Ypma et al. (2020), none of the surface floats appears to enter the Barents Sea).

In the absence of direct topographic guidance, the flow from the NwAFC towards the NwASC/BSO is bound to be slow and meandering, likely goverened by mesoscale eddy activity. This in turn suggests high susceptibility to both vertical heat fluxes and lateral mixing, making it likely that these waters would be colder and fresher than those flowing directly in the NwASC.

The reviewer states that "the reported advection time lags of up to 12 months between the relatively closely positioned GI and BSO likely involve this outer current branch". However, in agreement with existing literature, we argue that internal processes within the current such as shear and mesoscale meanders may explain large ranges of advection speeds that are up to a factor of 10 slower than the mean current without invoking interaction with different current branches. On a distance of 700 km, this makes for advection speed of about 2 cm/s, which would correspond to a mean current of about 15-20cm/s (Årthun et al. 2017), which seems reasonable as average between the vigorous Gimsøy and sluggish BSO currents.

In summary, we believe that flow from the NwAFC into the BSO is not a major factor we need to consider in this manuscript. Nevertheless, we agree that it is important information that needs to be conveyed transparently in the manuscript.

**Changes in the manuscript:**

- We now mention the possible interaction between NwASC and NwAFC in the introduction (l.28-31): "In a time-mean Eulerian sense, the two branches are well-defined and separated until reaching Fram Strait (e.g. Huang et al., 2023). However, recent studies based on Lagrangian tracking of drifters and simulated particles suggest some interaction between these branches, likely driven by mesoscale eddy activity in the eddy-rich Lofoten Basin region (Ypma et al., 2020; Broomé et al., 2021)."
- We also argue for the robustness of our salinity-based core definition in the BSO against possible NwAFC influence in the method section (I.145-148): "This yields a core consisting of the highest salinity waters, thus ensuring that it is indeed waters from the NwASC core (figure 3c). This approach excludes fresher coastal current waters as well as waters originating from the NwAFC that may have recirculated towards the BSO (Broomé et al., 2021), but most likely would have freshened (and cooled) substantially en route."
- In terms of the narrowing and broadening of the AW extent in response to wind forcing (NAO), as discussed in Blindheim et al. (2000), we added a reference in the discussion (I.333-339): "Using hydrographic data spanning both the NwASC and NwAFC north of Svinøy, Blindheim et al. (2000) found a broadening (narrowing) of the AW expanse in response to decreased (increased) wind forcing that has been observed to correlate with NAO on

inter-annual time scales. Experiments with varying AW core at Gimsøy (following the same definition as described in section 2.1.2, but estimated at every time step instead of based on the mean fields) showed indeed some (intermittent) anti-correlation of core width with NAO, but no systematic pattern of core width changes in the periods of marked discrepancies between modeled and observed anomalies (e.g., 2000–2005 and post-2017)."

Årthun, M., Eldevik, T., Viste, E., Drange, H., Furevik, T., Johnson, H. L., & Keenlyside, N. S. (2017). Skillful prediction of northern climate provided by the ocean. *Nature Communications*, *8*. https://doi.org/10.1038/ncomms15875

Ypma, S. L., Georgiou, S., Dugstad, J. S., Pietrzak, J. D., & Katsman, C. A. (2020). Pathways and Water Mass Transformation Along and Across the Mohn-Knipovich Ridge in the Nordic Seas. *Journal of Geophysical Research: Oceans, 125*(9). <a href="https://doi.org/10.1029/2020JC016075">https://doi.org/10.1029/2020JC016075</a>

Broomé, S., Chafik, L., & Nilsson, J. (2021). A Satellite-Based Lagrangian Perspective on Atlantic Water Fractionation Between Arctic Gateways. *Journal of Geophysical Research: Oceans*, *126*(11). https://doi.org/10.1029/2021JC017248

Huang, J., Pickart, R.S., Chen, Z. *et al.* Role of air-sea heat flux on the transformation of Atlantic Water encircling the Nordic Seas. *Nat Commun* **14**, 141 (2023). <a href="https://doi.org/10.1038/s41467-023-35889-3">https://doi.org/10.1038/s41467-023-35889-3</a>

Ensure that your Conclusions clearly state that you have identified: 1) reduced cooling north of Gimsøy and 2) established a link between this phenomenon and reduced sea-to-air heat fluxes. Additionally, 3) with a more realistic inclusion of the outer baroclinic currents and your (trimmed) advection-lag analysis, you could also explain the 2000-2005 and post-2017 periods. You should mention eddy fluxes, but in a more tentative manner.

We slightly modified the wording in the conclusions to emphasize that surface heat fluxes are likely the dominant driver, in conjunction with increased advection. The formulation reflects our view on the main results.

**Details**

Lines 9-11: This wording may give the impression of temporal cooling. Please use better wording, similar to the description at the beginning of the Introduction (lines 21-24)

We changed the wording to: "Atlantic Water (AW) cools and freshens during its journey along the Norwegian coast, but our analysis shows that north of Lofoten (69° N), the cooling has reduced by 0.11-0.13 ° C /decade."

L27: "... while the more baroclinic NwAFC..."

Changed as proposed.

L28: southern Norwegian location?

Changed to "At Svinøy (off southwestern Norway ~69°N)"

L34: bifurcates

Corrected

L37: to à toward

Corrected misspelling of the word "northwards" and changed "to" into "towards".

L40: effect à affect

**Corrected**

L81: Why do you only include data up to 2022

When starting the analysis this was the common time span available for all datasets. It also means that since we go to the end of 2022, we have exactly 30 years of data.

L214: Delete 'figure'

**Corrected**

L315-323: The year 2003 was characterized by very weak and broad boundary current system. This condition likely both reduced the advection speed, and increased the surface area of the AW ( $B_{surf}$ ). Inclusion of a temporally variable  $B_{surf}$  could thus help you explain the anomalous signals around both 2003 (large  $B_{surf}$ ) and 2017 (smaller  $B_{surf}$ ).

Indeed the transport at Gimsøy was very weak in 2003 (figure 3j). And a variable surface area would affect the Box model in the same (linear) way as changes in advection speed. However, since we did not find any systematic pattern of AW core width changes at Gimsøy during 2003, we opted not to include this complication in the box model. Instead, as mentioned above, we refer to this topic (and Blindheim et al., 2000) in the discussion (I.333-339):

"Using hydrographic data spanning both the NwASC and NwAFC north of Svinøy, Blindheim et al. (2000) found a broadening (narrowing) of the AW expanse in response to decreased (increased) wind forcing that has been observed to correlate with NAO on inter-annual time scales. Experiments with varying AW core at Gimsøy (following the same definition as described in section 2.1.2, but estimated at every time step instead of based on the mean fields) showed indeed some (intermittent) anti-correlation of core width with NAO, but no systematic pattern of core width changes in the periods of marked discrepancies between modeled and observed anomalies (e.g., 2000–2005 and post-2017)."

**Figures**

**Figure 1**

Just out of curiosity, what do the negative (southward, blue) flows signify? There appear to be southward flows between the poleward AW current branches, with the highest values at the foot of the slop (flanking the slope current). Are these real, or artifacts from your combined altimetry-hydrography analysis?

We thank the reviewer for bringing this issue to our attention. We added the following statement/expalantion in the methods section (starting in line 119):

"We note that in the time-averaged AGV sections, there appears to be a weak southward flow underneath the NwASC current core at Si, GI and BI (figure 1d, j, m). Due to the slope of the isopycnals, relative geostrophic currents are all southward in these sections, typically increasingly so with depth. The northward flow only comes to be when adding the satellite-derived surface geostrophic currents. It is thus possible that these southward flows underneath the current core may be artifacts, either due to slightly too steep isopycnal gradients at the slope, or too weak surface geostrophic northward currents (owing to the spatial averaging involved in the satellite product processing). Since overall transport estimates (presented in section 2.1.2) are within the ranges reported in literature, we do not consider this to be problematic for our analyses."

Concerning the weak southward flow at GI offshore of the NwAC core, this is due to negative surface geostrophic currents (density gradients are very small there) and is not an artifact of our data processing. The same can be seen in ship-based ADCP measurements.

Figure 7

Omit panel a)

**Removed as suggested.**

**Figure 8**

Why not show the actual temperature differences between GI and the BSO,  $T_{GI}$  -  $T_{BSO}$ ? These must, clearly, always positive (Figure 4). As presented, an incautious reader might interpret Figure 8 as indicating warming from GI to the BSO, after around 2011.

The reviewer is right in that the temperature difference is always positive. The reason we do not show absolute values is that - in the absence of proper mixing parameterizations - the box-model estimates are off by an unknown factor and only make sense as anomalies. Thus, to make the curves comparable we chose to show both as anomalies. We note this in lines 200-202.

**Reviewer #2**

The paper is interesting, carrying an analysis of a massive series of repeated sections across the Norwegian Atlantic sloope Current, and the results seem to have been thoroughly investigated and supported.

I nonetheless have a series of comments and questions, that may be helpful to better grasp the main results.:

We thank the reviewer for their assessment of our work. The comments and questions have prompted us to improve and clarify our manuscript.

In AW current core for time series the top 50 m are removed (and thus also Ekman...), as well as constrain on maximum bathymetric depth (for SI and GI), in addition to constrain on S 35.00 (for practical salinity; for SA 35.16). What justifies not taking the top 50 meters? (or did I mis-understood?)

You understand correctly. The reasoning behind the exclusion of the top 50 m is that during summers, there can be a very warm SML. This could introduce a (slight) seasonality in the time series that is not really related to AW properties, but purely local. Tests showed that the main results are robust whether we remove this layer or not; we deem it cleaner to have it removed. We clarified the wording in the manuscript (l.133) to "Has an upper boundary of 50 m below sea surface (to exclude large parts of the seasonally varying surface layer)".

Then, the authors separate the core and non-core parts of this transport (based either on largest current or largest S, depending of the section to 50% of the transport...). I am not exactly sure of the practical interest to make this separation

(except after to show that the results are someway independent on the part taken?). Upstream AW properties change, so what? Maybe it could be more straightforward to show the core+non-core, and then say (or in an appendix) that results are results hold in core (and non-core) that for the whole core+non-core (is that correct? I don't think that this is shown or really discussed; indeed, in the abstract, it seems that there is more emphasis on what happens in the core, but after in the presentation, there is often a symmetry of results presented core and non-core)

Afterwards, even when describing the average core and non-core properties, one sees how each criteria (currents, bathymetry, salinity) strongly constrains the spatial extent of core and non-core; It seems to me that this might complicate a bit complicated the comparison of variations in different sections (in particular, between Svinoy and Gimsoy).

The reasoning behind the split in core/non-core areas is that the sections have different lenghts and the fraction of the section occupied by the NwAC is also different. This is somewhat ameliorated by choosing fix isobaths as outer limit for the SI and GI sections, but these cutoffs are rather crude and only meant to exclude currents that are not part of the NwASC system. We argue that our core definition makes the sections more comparable. We modified the wording in first sentence describing the core definition (I.128) to: "Obtaining meaningful and comparable time series of properties from different sections of varying spatial coverage of the NwASC can be non-trivial."

The main result (table 1 and figure 5) is that there is a decrease in the cooling between the sections off Norway and the northern sections, in addition to the multi-decadal variability found in all the time series (more for S than T, although in T I see it more in the non-core time series than in the core time series). The authors find reduced heat loss (sensible heat loss decreasing) and faster advection speed.

Question: what are uncertainties in the heat flux products and to which degree this can be taken at face value (maybe because it is on sensible component, and TA is better and more consistently reproduced in the reanalyses). I am not 100% sure that I fully understand the second result, which is not that clear to me in the time series from geostrophic velocity (and estimate of eddy transfer). Indeed, this interesting result is based not on the currents but on the lag correlations for salinity.

It is a good question about the uncertainties in the heat fluxes products. Unfortunately, ERA5 does not provide any estimates. However, Meyer et al. (2023) compare ERA5 heat flux trends in several regions, including the Norwegian Sea, with independent estimates from energy budget closure. While uncertainties are substantial, for the Norwegian Sea region they are confident that at least the direction of the trend (i.e. reduced heat fluxes from the ocean to the atmosphere) is robust.

We added this to the description of the heat fluxes in the data/methods section in l. 185-187.

Mayer, J., Haimberger, L., & Mayer, M. (2023). A quantitative assessment of air-sea heat flux trends from ERA5 since 1950 in the North Atlantic basin. *Earth System Dynamics*, *14*(5), 1085–1105. https://doi.org/10.5194/esd-14-1085-2023

On Fig. 7, I did not fully understand how the vertical shear in the core is estimated and how it is relevant. In some places, cores defined by velocity, so it is important to specify in which layer this vertical shear is estimated: is the layer constant in time or varying? Is it also averaged across the core? Depending on how it is defined, the interpretation of the change is different (negative shear also means higher northward velocity at depth: is that correct?)

Thank you for pointing this out. We added a sentence clarifying how the vertical shear is calculated (l.300): "To investigate this, we estimate vertical shear for each time step by creating a mean velocity profile from horizontally averaging AGV within the core and integrating the vertical shear of this profile."

The reason we introduce this estimate of vertical shear has to do with the fact that - as you note – there is no trend in either transports or geostrophic currents that might explain the changes we observe in salinity anomaly lags (figure 8). As we note in the introduction (l.64-70) and discussion (l. 348-352), other studies mention "internal" processes such as changes in shear as possible drivers of changing advection. Our estimates suggest, that there is a qualitative agreement that reduced advection time coincides with reduced vertical shear.

For interannual variability, figure 8 seems rather convincing (except for a little less than 10 years out of 31), with, if I understand correctly, advection maintained constant? I am no sure if trend removed? (I don't think so, but that was not so clear to me)

In this figure (new figure 6), we use different advection speeds (between 3 and 12 months) that are kept constant over time. The effect of the different speeds is represented as shading. The trend is not removed but only anomalies are shown. The reason we do not show absolute values is that -in the absence of proper mixing parameterizations-, the box-model estimates are off by an unknown factor and only make sense as anomalies. Thus, to make the curves comparable we chose to show both as anomalies. We mention this in the methods in lines 200-202.

**Minor comments**

1. 34 Bifurcate

**Corrected.**

2. 40: effect? Probably 'result in'

Changed to "effect".

3. 100: if I understand the approach correctly, smoothing just on depths of isopycnals on individual section (not isopycnal T and S). What is done close to the sea surface? (or when an isopycnal outcrops?)? or near the boundaries (when isopycnal section crosses bottom...)

Indeed, smoothing is done in terms of the depths of isopycnals. At the edges, data is by default reflected, such that the smoothing window size is maintained.

I understand what is done near bottom for extending T and S, and computing then Relative G V, but how is near-bottom transport then estimated? (I suspect taking exact bathymetry and multiplying by an area?)

This is a good point. The area around each AGV data point is assumed to be rectangular (mid-difference to the neighboring grid points). Generally, this will cause an overestimation of transport where the rectangular cells intersect with topography. In practice this is only a potential issue at Gimsøy, because the core area is away from topography at all other sections (figure 3). But even at Gimsøy this only affects a few individual cells.

We added an explicit mention of this in the methods in lines 137-139.

And then, step of using at core value (0-error) the altimetry-derived velocity (l.113) to have absolute GV. This seems a reasonable approach. However, what are the errors (resolution; spatial smoothing of altimetric product? Is it similar to the one done on the isopycnal depths?)

The nominal resolution of the geostrophic velocity dataset is 0.125° x 0.125°. This corresponds to about 7.2km in meridional direction and around 2.7km (at 69°N) in zonal direction. The actual resolution is much coarser but this is not specified in the copernicus product. The only error estimate given is the theoretical mapping error, that is around 3-5 cm/s, but the physical interpretation is not straight forward. Using large parts of the same underlaying satellite altimetry dataset, Mork & Skagseth (2010) investigated errors in the surface geostrophic velocities at the Svinøy transect (see their appendix). They found the velocity errors to be noticeable, but small compared to both spatial variability along the section as well as temporal variability on seasonal to inter-annual time scales (their figures 3, 4, 5 and 8).

**We added the latter statement in the methods section in lines 112-115.**

1. 204-209, I am wondering whether it would be better to present the trends difference Bjornoya-Svinoy an BSO-Svinoy, and not the other way around. Well, both can be argued... (saying than north warms more than south, thus less heat loss or other heat mechanism, seems to me a little bit easier to grasp, as water flows from south to north).

Indeed, both can be argued for. Our thinking is that cooling along the Norwegian continental slope is a given. Now we observe that there is less cooling than before, so a trend showing negative numbers makes sense. We therefore opt to maintain this way to calculate the differences.

Heat fluxes, first I. 150, but after when the plots are shown, I would adopt the ocean convention as heat flux towards the ocean. Here seems that it is heat flux towards the atmosphere (as seen in figures). Thus, with the convention adopted here, an ocean heat loss is associated with a positive heat flux. I prefer the opposite, as is the vast majority of oceanographic papers that I looked at to verify.

We understand that there can be confusion with different conventions. We did in fact originally work with the convention as suggested, but we found that since we are mainly concerned with trends, this would mean that reduced surface heat loss would be associated with positive trends in figure 7b and c. We found the notion that a reduction is associated with a positive trend more confusing and thus opted to adopt the convention that positive heat fluxes are directed upward. This give a negative trend for decreasing heat fluxes.

1. 258: '(Figure 7a,b)

Corrected.

---

## Author Response (AR2)

We again thank Hjálmar Hátún and the anonymous reviewer. Here we detail our responses (in blue) to the comments by the reviewers (black). Line numbers given by us refer to the revised manuscript (not the tracked-changes version).

**Reviewer #1**

We thank Hjálmar Hátún for the positive reaction to our response and related revisions and are pleased to find that there are no further requests to modify the manuscript.

**Reviewer #2**

I thank the authors for having proerly answered some of my comments/queries. On the other hand, I dont find that there was much effort to shorten/simplify the paper.

In particular, I dont fully understand the response brought to my comment that there does not seem
to be much added value in separating core and non-core properties, provided that their time series
are rather similar and that for the discussion on the transport (or the heat fluxes), one cannot really estimate separately the impacts on each of those. There are many other assumptions (on the extent of each domain, or, as the other reviewer points out, the inlfuence of what happens in terms of properties and transport west of the NASC), which, I believe, make the distinction adopted rather unimportant. I had suggested presenting and discusing one single parameter (instead of the two) (maybe some weighted average of the two sets of properties), leaving to an appendix, the separate presentation of the two. This has not been done/replied to. How do you justify it?

Despite that, I am still favorable for the publication of the paper.

We thank the reviewer for their overall positive sentiment towards our manuscript.

We acknowledge that the reviewer is not convinced about the split we do between the core and non-core areas.

We we have addressed the reasoning behind the split in our previous response and in the manuscript (lines 128-130): We split the sections in the core and non-core areas to make it possible to compare the NwASC core properties between sections because the sections are of different lengths.

Concerning the presentation of both core and non-core properties in the beginning of the results (sections 3.1 and 3.2 with related figures 4 and 5): Because definining a core area is somewhat arbitrary, we feel that it is important to illustrate that we indeed capture the warm and salty AW core reliably (as seen in figures 4 and 5), and that the result is robust and not sensitive to the exact choice of cutoff between core and non-core (the time series are highly correlated, figure 5a-h). We mention the latter in lines 151-153 in the manuscript.

In the remainder of the manuscript, including all further analyses and discussions, we only use the core properties, which is in agreement with the reviewer's request to only use a single property.

We now emphasize this by adding the word "core" in the heading of 3.3. and adding the following statement to the end of section 3.2 (line 225-226): "All analyses and discussions in the remainder of this manuscript are based on the core properties only."